# Deletion of the *Candida albicans TLO* gene family results in alterations in membrane sterol composition and fluconazole tolerance

James O'Connor-Moneley[1], Jessica Fletcher[1¤a], Cody Bean[1], Josie Parker[2¤b], Steven L. Kelly[2], Gary P. Moran[1☯*], Derek J. Sullivan[1☯*]

**1** Division of Oral Biosciences, Dublin Dental University Hospital, and School of Dental Science, Trinity College Dublin, Dublin, Ireland, **2** Institute of Life Science, Singleton Campus, Swansea University, Swansea, Wales, United Kingdom

☯ These authors contributed equally to this work.
¤a Current address: Department of Integrative Biology, University of Colorado, Denver, CO, United States of America
¤b Current address: School of Biosciences, Cardiff University, Cardiff, Wales, United Kingdom
* derek.sullivan@dental.tcd.ie (DJS); gary.moran@dental.tcd.ie (GPM)

**Data Availability Statement:** All RNA sequencing data files are available from the NCBI sequence read archive database (BioProject no.

## Abstract

Development of resistance and tolerance to antifungal drugs in *Candida albicans* can compromise treatment of infections caused by this pathogenic yeast species. The uniquely expanded *C. albicans TLO* gene family is comprised of 14 paralogous genes which encode Med2, a subunit of the multiprotein Mediator complex which is involved in the global control of transcription. This study investigates the acquisition of fluconazole tolerance in a mutant in which the entire *TLO* gene family has been deleted. This phenotype was reversed to varying degrees upon reintroduction of representative members of the alpha- and beta-*TLO* clades (i.e. *TLO1* and *TLO2*), but not by *TLO11*, a gamma-clade representative. Comparative RNA sequencing analysis revealed changes in the expression of genes involved in a range of cellular functions, including ergosterol biosynthesis, mitochondrial function, and redox homeostasis. This was supported by the results of mass spectrometry analysis, which revealed alterations in sterol composition of the mutant cell membrane. Our data suggest that members of the *C. albicans TLO* gene family are involved in the control of ergosterol biosynthesis and mitochondrial function and may play a role in the responses of *C. albicans* to azole antifungal agents.

## Introduction

*Candida albicans* is widely recognized as one of the most important fungal pathogens of humans, causing infections in a variety of anatomic locations, including the oral cavity, vagina, skin, and the bloodstream [1–3]. This pathogenic yeast species has recently been included in the "Critical Group" of the World Health Organization's fungal priority pathogens list [4]. Although *C. albicans* is a minor component of the commensal microbiota, given the opportunity it can become an opportunistic pathogen that can cause life threatening illness,

PRJNA1129456); https://www.ncbi.nlm.nih.gov/bioproject/?term=PRJNA1129456.

**Funding:** DJS - and GPM - Science Foundation Ireland 19/FFP/6422 www.sfi.ie The funders did not play any role in study design, data collection, analysis or in the decision to publish.

**Competing interests:** The authors have declared that no competing interests exist.

particularly in immunocompromised patients. In the USA, *Candida* species, most predominantly *C. albicans*, are the ninth most common cause of bloodstream infection, with an estimated mortality rate of 20% of reported cases [5, 6]. Treatment options for fungal infections are relatively limited compared to those for bacterial infections, with the fungistatic triazole fluconazole being the drug used most frequently in the treatment of candidiasis [7]. Treatment is further complicated by reduced drug efficacy due to resistance and tolerance mechanisms that can develop in *C. albicans* when exposed to fluconazole [5, 8].

The primary target of azole drugs is Erg11, a sterol C14-demethylase, which is an enzyme in the ergosterol biosynthetic pathway [9]. Ergosterol is an essential component of and the most abundant sterol in fungal cell membranes, where it plays a role similar to cholesterol in human cell membranes [9]. Common azole drug resistance mechanisms in *C. albicans* include point mutations in the Erg11 protein that reduce azole drug affinity, increased expression of the *ERG11* gene, formation of protective biofilms and increased drug efflux due to increased expression of genes encoding transporter proteins such as Cdr1 and Mdr1 [5, 8]. In addition to antifungal resistance, antifungal tolerance has been increasingly recognized as another contributing factor to reduced efficacy of antifungal drugs, especially to triazoles such as fluconazole. Antifungal tolerance differs from resistance in that it is a transient stress induced state, which the affected progenitor cells do not pass to their descendants via heritable genetic mutations [10–13]. The genetic changes that give rise to antifungal resistance lead to stable increases in the Minimum Inhibitory Concentration (MIC) of the drug, while strains with non-inherited tolerance display the ability to persist in the presence of the drug, but have an unaltered MIC compared to wild type cells [14]. For this reason, tolerance is usually identified as a subpopulation of cells that grow above the MIC when cultured for periods longer than the 24 hours routinely used to measure antifungal susceptibility [10]. It has been suggested that this difference may account for the discrepancy between the reported proportion of resistant isolates recovered from *C. albicans* infections, which is less than 1%, and the relatively high levels of treatment failure [12].

As the clinical importance of azole tolerance is more widely recognized, we are beginning to improve our understanding of the molecular basis for this phenotype and recently it has been suggested that global stress responses are key to its development [12, 13]. Studies in *Candida glabrata* and *Saccharomyces cerevisiae* have shown that mitochondrial function and activity have been associated with responses to antifungal drugs. In particular, alteration of mitochondrial activity can lead to changes in membrane composition and permeability [15–18]. It has been shown previously that activation of pleiotropic drug resistance (PDR) signalling, resulting from mitochondrial dysfunction, regulates the expression of genes required for homeostasis of phospholipids and sphingolipids, two key components of the plasma membrane [19, 20]. It has also been suggested that drug induced changes to plasma membrane and cell wall composition, leading to the development of antifungal tolerance, are regulated by the ubiquitous molecular chaperone Hsp90 [21]. This protein plays a central role in stress responses and initiates two distinct signalling cascades, the calcineurin and protein kinase A (PKA) pathways, which direct changes to plasma membrane and cell wall integrity, respectively. Inhibition of the calcineurin pathway, which controls calcium homeostasis, with the immunosuppressant drug cyclosporine A has been shown previously to inhibit the development of fluconazole tolerance in *C. albicans* [10, 21].

In the present study, we describe the fluconazole tolerance phenotype exhibited by a recently described *Δtlo* mutant of *C. albicans* in which all members of the greatly expanded TeLOmer associated (*TLO*) gene family (n = 10–15) have been deleted [22]. *TLO* genes encode the Tlo/Med2 subunit of Mediator, which is a multi-subunit complex of polypeptides involved in the regulation of transcription by acting as an intermediary between regulatory sequences and DNA-bound activators or repressors with the basal transcriptional machinery (RNA

Polymerase II) in all eukaryotes [23, 24]. The *TLO* gene family in *C. albicans* is comprised of three distinct clades. In our previous studies [22], representatives of the alpha- and beta-clades, were able to complement almost all mutant phenotypes. However, two gamma-clade representatives (i.e. *TLOγ5* and *TLOγ11)*, were found to be unable to complement the mutant. The role of various Mediator subunits in the responses to antifungal agents has been investigated in various fungal pathogens, including *Candida* species and in *A. fumigatus* [25–28]. In *C. albicans* the Mediator tail subunit Med3 and a subunit of the kinase domain CaSNN3 were shown to be involved in Tac1-mediated *CDR1* drug efflux pump expression [26, 27]. The current study characterises the responses to fluconazole of the *Δtlo* mutant and investigates associated alterations in metabolic pathways and membrane composition and their potential roles in mediating azole tolerance.

## Materials and methods

### Strains and growth conditions

The strains used in this work are listed in S1 Table. All *C. albicans* strains, including the WT isolate MAY1244 and the *Δtlo* mutant, used in this study were derived from the laboratory strain SC5314. *Candida* strains were routinely grown in Yeast Extract Peptone Dextrose (YEPD) medium (liquid or solid) at 37˚C. *E. coli* strains were grown in Lysogeny Broth (LB) medium at 37˚C. RPMI-1640 liquid medium with added glucose (2%), MOPS (160 mM) and leucine (0.13%), herein referred to as "complete" RPMI liquid medium, was used for fluconazole sensitivity assays and mitochondrial activity assays. The *Δtlo* mutant, in which all 14 *TLO* genes have been deleted was generated using CRISPR-Cas9 mutagenesis, and clade-representative reintegrants were described previously by Fletcher *et al.* [22]. Additional overexpression mutants were generated in the *Δtlo* mutant background using the method described by Milne *et al.* [29], whereby the enolase promoter was inserted in front of individual genes of interest; i.e. *ERG1*, *ERG3*, *ERG6*, *ERG11*, *ERG251*, and *UPC2*.

### Growth rate assays

Growth rate analysis was carried out in YEPD at 37˚C. Cultures were standardized to an $OD_{600}$ of 0.1 in 25 ml (inoculated from an overnight culture) and the $OD_{600}$ was measured every hour. For fluconazole microbroth dilution assays, overnight cultures were grown from single colonies in 4 ml YEPD at 37˚C with 200 rpm shaking, washed in sterile PBS and resuspended to density of $2 \times 10^6$ cells per ml. Five μl of the equalized culture was used to inoculate a 96-well microtiter plate in which a gradient of fluconazole concentrations were prepared by serial dilution in RPMI liquid medium. The microtiter plate was incubated statically for 24hrs at 37˚C and the absorbance read at 600nm on a FLUOstar Omega microplate reader (BMG Labtech). Data were analysed using Microsoft excel and graphs generated on GraphPad Prism (v9).

### Antifungal susceptibility testing

Fluconazole disk diffusion assays were prepared using solid RPMI medium. Overnight cultures were grown from a single colony in 4 ml YEPD at 37˚C with 200 rpm shaking, washed with sterile PBS to remove YEPD medium, and resuspended to $OD_{600}$ = 0.07. Sterile cotton swabs were used to streak the diluted cultures onto solid RPMI agar plates and the plates allowed to air dry for 15 mins. A 25 μg fluconazole susceptibility disk (OXOID), or fluconazole Etest (bioMérieux), was placed at the centre of each RPMI agar plate and incubated at 37˚C for 48 h. The plates were imaged on a "FLASH & GO" plate imager (IUL Instruments) using auto-exposure settings. The images were then subsequently analysed using the diskImageR

pipeline31 and the R script available at https://github.com/acgerstein/diskImageR/blob/master/inst/walkthrough.R in order to assess the Fraction of Growth (FoG) inside the zone of inhibition and the Radius (RAD) of clearance [30].

## RNA-sequencing and analysis

Overnight cultures of the wild type strain May1244 and the *Δtlo* mutant were grown from single colonies in 4 ml YEPD at 37˚C with 200 rpm shaking. The cells were washed in sterile PBS and resuspended to $OD_{600}$ = 0.07. Sterile cotton swabs were used to streak the diluted cultures onto solid RPMI agar plates. Plates were incubated for 48 h at 37˚C on solid RPMI medium with and without a fluconazole Etest. Cells were harvested from the zone of inhibition surrounding the Etest strip by washing with sterile PBS, cells were also taken from the corresponding zone on the plate without the Etest strip. RNA was then extracted using the RNeasy extraction kit (QIAGEN) per the manufacturer's instructions. mRNA sequencing was performed with strand-specific libraries and sequenced on the Illumina Nova-Seq 6000 Sequencing System using paired end 150 bp reads. Each experiment generated a minimum > 20 million read pairs per sample with Q30 score $\geq$ 85%. Raw reads were aligned to the *C. albicans* SC5314 Assembly 22 genome (downloaded from the *Candida* Genome database (CGD) [31]) in the Strand NGS4.0 software package using the default settings. Reads were quantified and normalized in StrandNGS using DeSeq2 [32] and statistical analysis of differential gene expression was carried out with post-hoc Benjamini-Hochberg testing performed by default (FDR q < 0.05). Further analysis on lists of differentially expressed genes was performed via GO analysis on the *Candida* Genome Database [31, 33, 34] and GSEA [35]. Sequence data is available for download from the NCBI sequence read archive, BioProject no. PRJNA1129456.

## Quantitative RT-PCR

Expression levels of specific genes and constructs were determined by qRT-PCR. cDNA was generated from RNA extracted from strains using the RNeasy kit (Qiagen). Specific primers for qRT-PCR are listed in S2 Table. qRT-PCR was carried out in biological triplicate using an Applied Biosystem 7500 Fast Real Time PCR System, with the expression of *ACT1* used as an endogenous control. Differential expression was calculated as described by previously [36] and the results graphed in GraphPad Prism (v9).

## Sterol composition analysis

The total cell sterol content of selected *C. albicans* strains and mutants was determined as described previously [37, 38]. Briefly, the yeast cells were grown from single colonies in 10 ml YEPD at 37˚C with 200 rpm shaking. Subcultures were prepared in 50 ml YEPD to an $OD_{600}$ = 0.1 and cultured until they had reached mid-exponential phase ($OD_{600}$ = 0.6–0.8), both in the presence and absence of sub-inhibitory concentrations of fluconazole (0.25 µg/ml). Cells were harvested and non-saponifiable lipids were extracted, dried in a vacuum centrifuge, and were derivatized with trimethylsilane (TMS). The TMS-derivatized sterols were analyzed by gas chromatography-mass spectrometry (GC-MS). The GC-MS data files were analyzed, and results of three replicates from each sample were used to calculate the mean percentage 6 standard deviation for each sterol [37, 38].

## Mitochondrial activity assays

Mitochondrial activity of the wild type, *Δtlo* and *Δmed3* mutants, and *TLO* reintegrants was assessed by measuring the oxidative consumption rate (OCR) by use of the MitoXpress Xtra

Oxygen Consumption Assay Kit (Agilent). Instructions were carried out as per the manufacturer's instructions. Briefly, the cells were cultured over night from single colonies in 4 ml YEPD at 37˚C with 200 rpm shaking. The cells were washed twice in sterile PBS and resuspended to an $OD_{600} = 1.52$. A 96 well microtiter plate was prepared in which each well contained 90 μl of RPMI medium plus 10μl of MitoExpress reagent. The plate was split into three sections; the first containing RPMI medium without any additional additives, the second containing 0.2 5μg/ml fluconazole, and the third containing 10 μg/ml cyclosporine. The plate was then preincubated at 37˚C for 30 mins before inoculation with 12.5μl of the washed and equalized cultures. Each well was then sealed with a drop of oil and resolved fluorescence was read at Ex 340 ± 50 nm / Em 655 ± 25 nm on a FLUOstar Omega microplate reader for a total of 6 hours. The lifetime slope of range (2–4 hours) was used to calculate the OCR and results were plotted using GraphPad Prism (v9).

### Efflux of rhodamine 6G as measurement of drug efflux capacity

Measurement of Rhodamine 6G (R6G) efflux was carried out as described previously [39] with slight modifications. In brief, yeast cells were cultured from a single colony of each isolate was suspended in 5 ml Synthetic Dextrose (SD), (6.7 g Bacto Yeast Nitrogen Base without amino acids and 20 g glucose/litre) medium with 2% glucose and incubated at 37˚C with shaking at 180 rpm. The following day, 0.1 OD of cells were exponentially grown (6–7 h) in SD medium with 2% glucose at 37˚C with shaking at 180 rpm. $10^8$ exponentially growing cells from each isolate were then washed three times with 1 × phosphate-buffered saline (PBS; pH 7.4), starved for 2 h in 1× PBS containing 5mM deoxy-glucose, and incubated at 37˚C. Five μl of a 10 mM stock of R6G was added to 5 ml of a starved cell suspension to a final R6G concentration of 10 μM. The 10mM stock solution of R6G was prepared in 100% ethanol. The cells were then incubated at 37˚C for 30 mins. The cells were then pelleted and washed once with 1X PBS. The washed cells were resuspended in 1× PBS plus 2% glucose to initiate efflux. 200 μl samples were then collected at 0, 30, and 60 min intervals. The collected samples were pelleted, and the absorbance of the supernatants read at 527 nm.

## Results

### Deletion of *TLO* genes reduces sensitivity of *C. albicans* to fluconazole

We have previously shown that deletion of the entire *TLO* family in *C. albicans* affects a wide variety of cellular processes [22]. In this study, we wanted to investigate if the deletion of the gene family also affects responses to fluconazole, therefore we measured the fluconazole susceptibility of the wild-type parental strain MAY1244, the *Δtlo C. albicans* mutant and representative reintegrant strains using fluconazole disk diffusion and Etest assays. Because azole tolerance is not evident after 24 h of growth the plates were incubated for 48 h. The results from these experiments show that the *Δtlo* mutant was clearly less sensitive to fluconazole compared to the parent strain, and that fluconazole susceptibility was partially restored upon the reintroduction of pENO-*TLOα1* and pENO-*TLOβ2*, however reintroduction of the pENO-*TLOγ11* gene had no detectable effect on susceptibility (Fig 1A). In order to quantify the differences between strains we measured the fraction of growth (FoG) from within the zones of inhibition in the disk diffusion plates. The results show that the FoG of the *Δtlo* mutant was significantly higher than the FoG of the WT strain, suggesting that the mutant is more tolerant of fluconazole. Reintegration of pENO-*TLOα1* and pENO-*TLOβ2* in the *Δtlo* background led to reductions in the levels of tolerance, especially in case of *TLOβ2*, while reintegration of *TLOγ11* had no effect on the FoG (Fig 1B). In order to discriminate between resistance and tolerance phenotypes we repeated the experiments in the presence of cyclosporine A

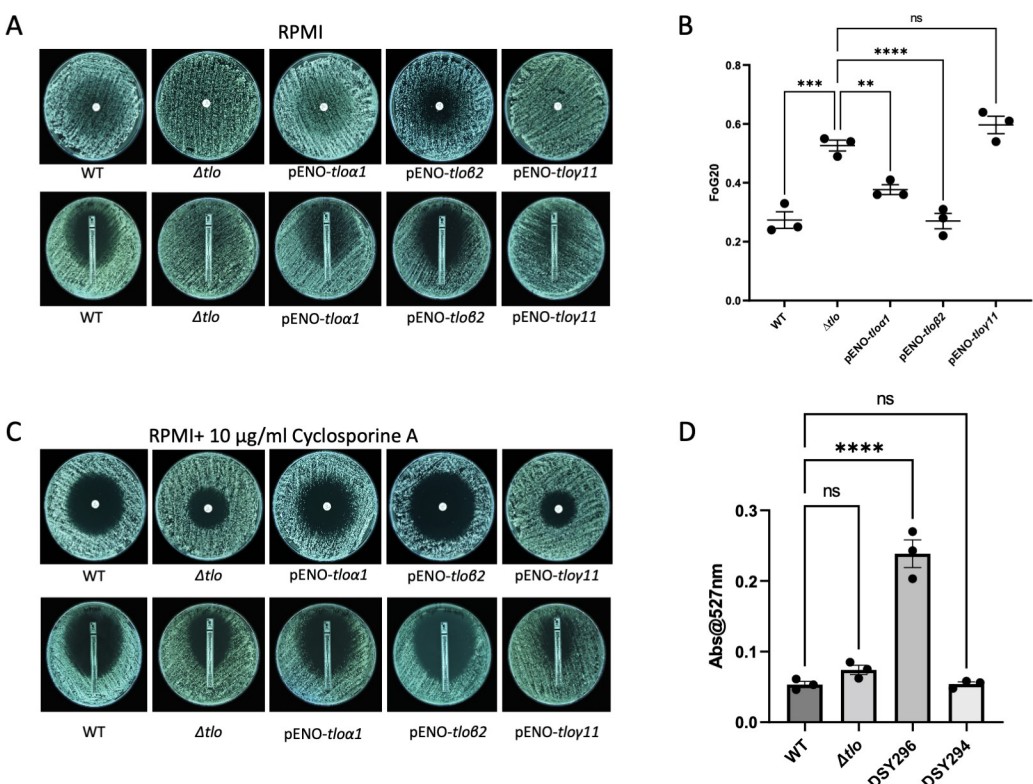

**Fig 1. Sensitivity assays comparing the levels of resistance and tolerance of the wild type strain, *Δtlo* mutant, and *TLO* reintegrants to fluconazole.** Fluconazole disk diffusion assays and fluconazole Etest assays comparing the sensitivity of WT, the *Δtlo* mutant, and the *TLO* reintegrants following a culture period of 48hrs at 37°C on RPMI agar (A). Fraction of Growth (FoG) measurements within the zones of inhibition in panel A, in the absence of cyclosporine A. Asterisks indicate statistically significant difference between the *Δtlo* mutant and other strains tested; ** = p-value < 0.01, *** = p-value < 0.001, **** = p-value < 0.0001 (B). Fluconazole disk diffusion assays and Fluconazole Etest assays comparing minimum inhibitory concentration (MIC) of the WT strain (0.25 μg/ml), the *Δtlo* mutant (0.75 μg/ml) and *TLO* reintegrants (pENO-*TLOα1*–0.19 μg/ml, pENO-*TLOβ2*–0.094 μg/ml and *TLOγ11*–1 μg/ml) following a culture period of 48 h at 37°C (C). Rhodamine 6G dye efflux assays comparing the active efflux of the WT strain, the *Δtlo* mutant to fluconazole resistant (DSY296) and fluconazole sensitive (DSY294) clinical isolates, asterisks indicate statistically significant difference in R6G dye efflux between the WT strain and fluconazole resistant clinical isolate DSY296; **** = p-value < 0.0001 (D).

(10 μg/ml), a calcineurin inhibitor which prevents the development of antifungal tolerance [10, 21]. This revealed that the *Δtlo* mutant, as well as the pENO-*TLOγ11* reintegrant, had zones of inhibition that were significantly smaller than those observed for the WT strain, the *TLOα1* reintegrant and *TLOβ2* reintegrant (Fig 1C). Etest assays were used to assess the differences in the minimum inhibitory concentration (MIC) of fluconazole between the WT strain, the *Δtlo* mutant and the *TLO* reintegrants, as the MIC could only be clearly assessed when fluconazole tolerance was inhibited in the presence of cyclosporine A (Fig 1C). The results revealed that compared to the WT strain, which had an MIC of 0.25 μg/ml, the MIC of the *Δtlo* mutant was 3-fold higher, with an MIC of 0.75 μg/ml; suggesting an increase in resistance. Reintegration of pENO-*TLOα1* and pENO-*TLOβ2* restored WT sensitivity, with MICs of 0.19 μg/ml and 0.094 μg/ml respectively, while reintegration of pENO-*TLOγ11* remained unchanged from the *Δtlo* mutant, with an MIC of 1 μg/ml (Fig 1C). To assess the involvement of active drug efflux in the observed differences in MIC between the WT strain and *Δtlo* mutant, rhodamine 6G dye efflux assays were performed. The efflux activity of the WT strain,

the *Δtlo* mutant and two reference clinical isolates (the fluconazole resistant isolate DSY296 and the fluconazole sensitive isolate DSY294 [40]) were compared (Fig 1D). The results show that there was an increase in rhodamine 6G dye efflux in the *Δtlo* mutant compared to the WT strain in the presence of fluconazole, however, this difference was not found to be significant (Fig 1D). Taken together, these data suggest that deletion of the *TLO* gene family significantly affects fluconazole tolerance in *C. albicans*, with additional effects on the baseline level of fluconazole susceptibility (a 3-fold rise in fluconazole MIC from 0.25 μg/ml to 0.75 μg/ml).

## Comparative analysis of RNA sequencing data of the WT and *Δtlo* mutant grown on RPMI medium

To investigate the role of the *TLO* gene family in *C. albicans* tolerance of fluconazole we initially compared global gene expression of the WT strain and *Δtlo* mutant when cultured statically at 37°C for 48h on solid RPMI medium in the absence of fluconazole (Fig 2A, S3 Table). Analysis of the transcriptomic data revealed that a total of 793 genes were differentially expressed; 292 genes were downregulated and 501 upregulated in the *Δtlo* mutant compared to the WT strain (Fig 2B). Gene set enrichment analysis (GSEA) of the differentially expressed gene list revealed several categories of genes that were enriched in the transcriptomes of either the WT strain or the *Δtlo* mutant (Fig 2C). Compared to the *Δtlo* mutant, the WT strain had

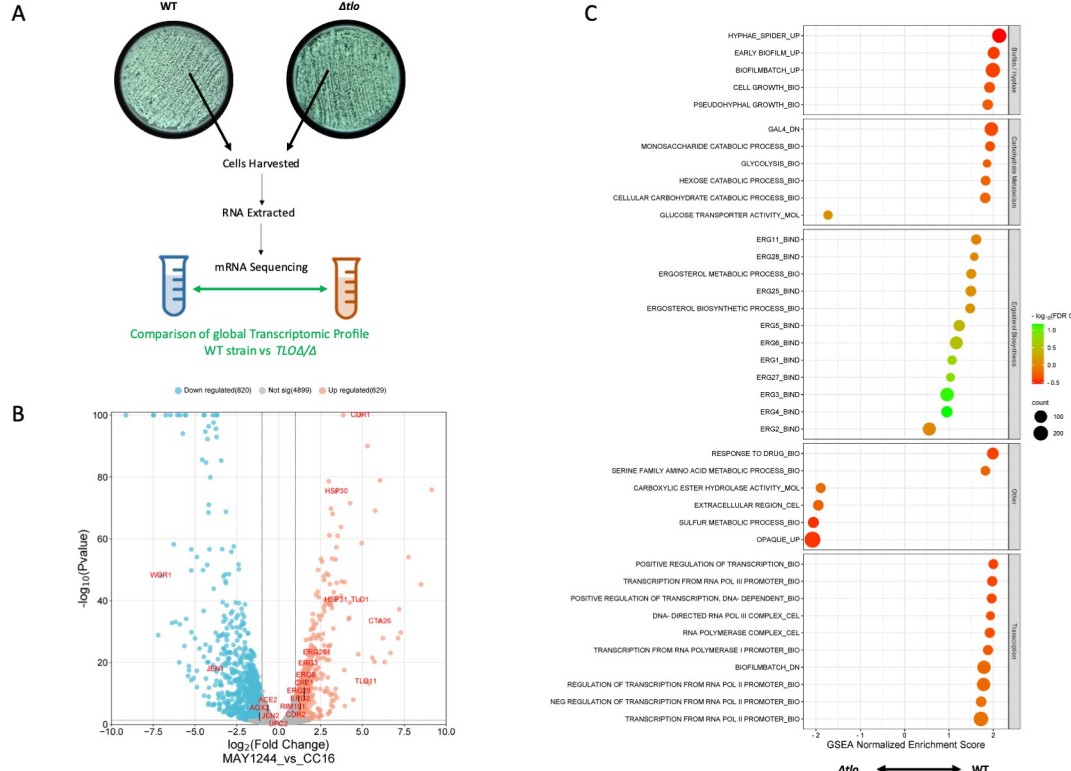

**Fig 2. Comparison of the transcriptomic profile of the WT strain and the *Δtlo* mutant grown on RPMI in the absence of fluconazole.** The WT strain and *Δtlo* mutant were cultured statically on solid RPMI agar plates for 48 h at 37°C, cells were harvested, RNA was isolated, and mRNA sequencing performed (A). Volcano plot showing the significantly differentially expressed genes between the WT and *Δtlo* mutant. Red dots indicate the genes that are more highly expressed in the WT strain compared to *Δtlo* mutant, while blue dots indicate the genes that are expressed significantly lower in the WT strain compared to the *Δtlo* mutant (B). Results of gene set enrichment analysis (GSEA) of the differentially expressed genes between the WT strain and the *Δtlo* mutant (C).

enriched expression of genes related to transcription, sterol biosynthesis, carbohydrate metabolism, biofilm formation and hyphal development (Fig 2C). Genes involved in opaque morphology, glucose transport, carboxylic hydrolyse activity and sulphur metabolism had lower levels of expression in the WT strain compared to the *Δtlo* mutant. These finding are similar to those described by Fletcher *et al.* which compared the transcriptomes of cultures grown in liquid nutrient-rich YPD medium at 37°C [22].

To identify the differences between the gene expression profiles of the WT strain and the *Δtlo* mutant when exposed to fluconazole, the global transcriptomic profiles of both WT and mutant cells growing in presence of fluconazole within the zone of inhibition on solid RPMI medium at 37°C were compared following incubation for 48hrs (Fig 3, S4 Table). A total of 587 genes were found to be differentially expressed between the *Δtlo* mutant and the WT strain; 400 genes were upregulated and 187 were downregulated in the *Δtlo* mutant compared to WT (Fig 3B). GSEA of the differentially expressed gene list revealed several categories of genes that were enriched in the transcriptomes of either the WT strain or the *Δtlo* mutant in the presence of fluconazole (Fig 3C). While sterol biosynthesis genes remained more enriched in the WT strain compared to the *Δtlo* mutant in the presence of fluconazole, it was not as significant an enrichment as in the absence of the drug. Other categories of genes that were

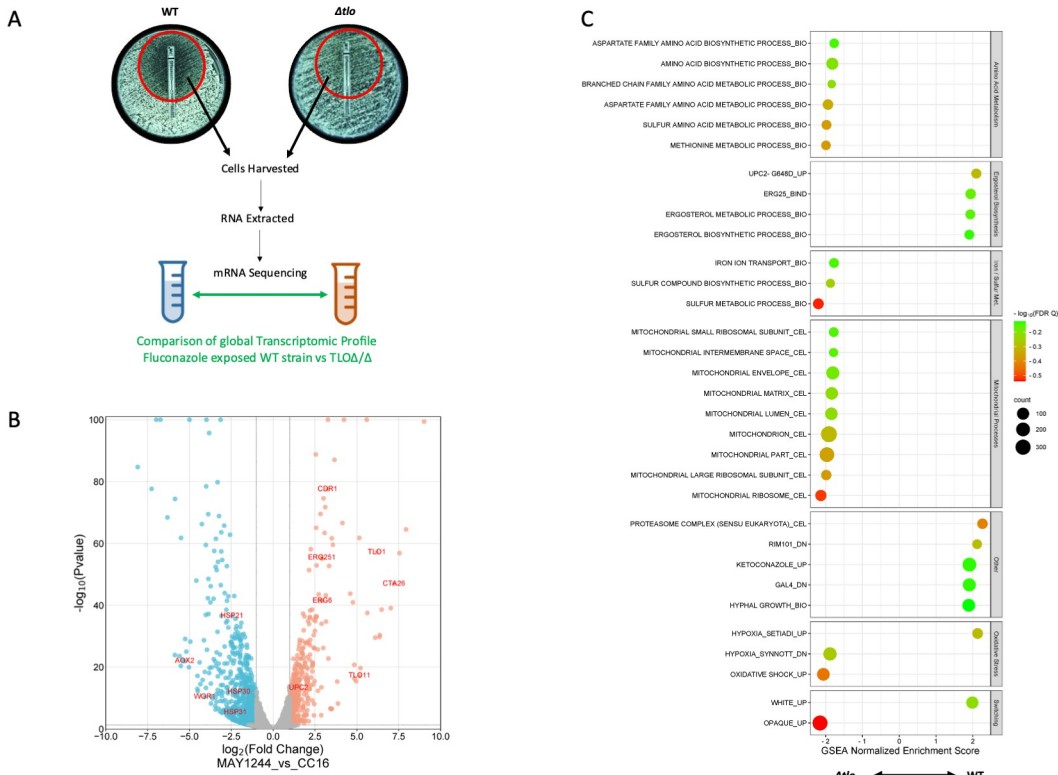

**Fig 3. Comparison of the transcriptomic profile of the WT strain and the *Δtlo* mutant in the presence of fluconazole.** The WT strain and *Δtlo* mutant were cultured statically on RPMI medium for 48 h at 37°C in the presence of a fluconazole Etest, cells were harvested from the zone of inhibition, RNA was isolated, and mRNA sequencing performed (A). Volcano plot showing the significantly differentially expressed genes between the WT strain and the *Δtlo* mutant. Red dots indicate the genes that are more highly expressed in the WT strain compared to the *Δtlo* mutant, while blue dots indicate the genes that are expressed significantly lower in the WT strain compared to the *Δtlo* mutant (B). Results of gene set enrichment analysis (GSEA) of the differentially expressed genes between the WT strain and the *Δtlo* mutant in the presence of fluconazole (C).

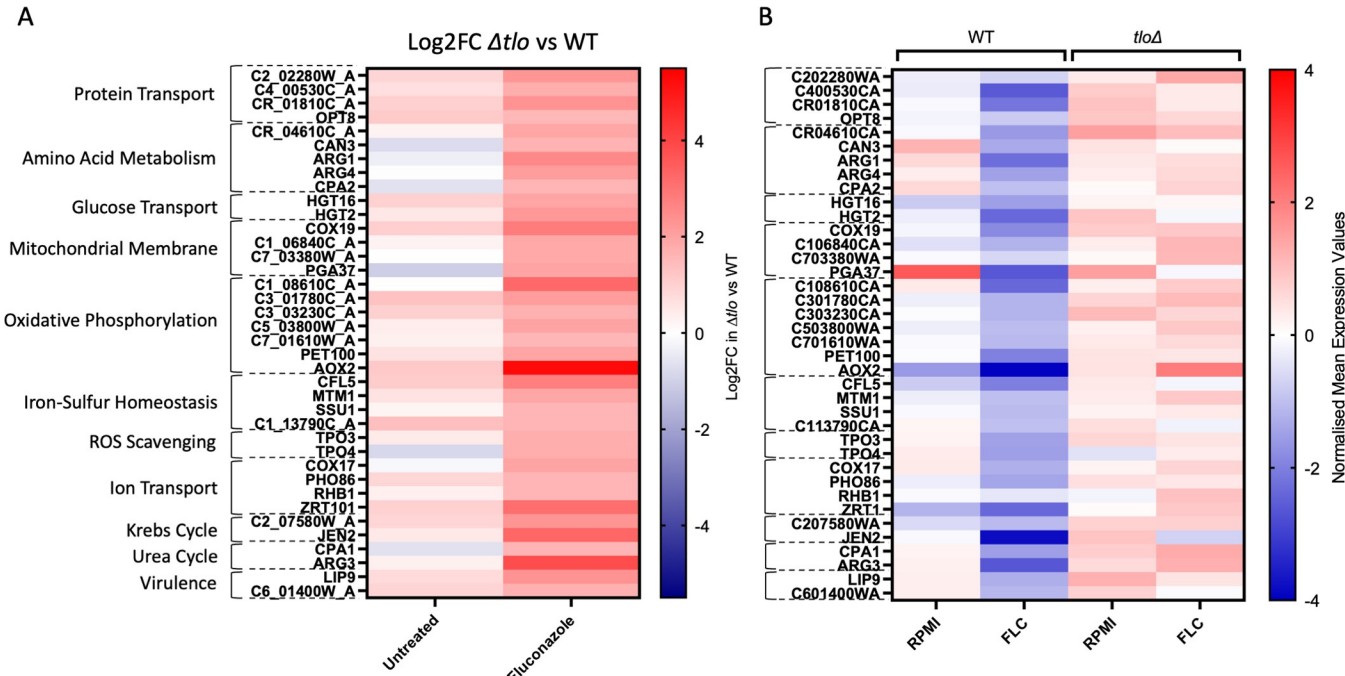

**Fig 4. Heatmaps showing the changes in the mRNA transcript levels of mitochondrial associated genes in the presence and absence of fluconazole.** Log2 fold changes in mRNA transcript levels of mitochondrial associated genes that were upregulated in *Δtlo* mutant compared to the WT strain in the presence and absence of fluconazole are shown (A). The normalized mean expression values across all samples are also shown (B). This comparison reveals that fluconazole induces a reduction in the expression of mitochondrial associated genes in the WT strain, while their expression in the *Δtlo* mutant is largely unchanged in the presence or absence of the drug.

enriched in the WT strain include genes upregulated in response to ketoconazole ("KETOCO-NAZOLE_UP"); genes downregulated upon deletion of carbohydrate metabolism regulator *Gal4* ("GAL4_DN"); and genes induced by hypoxia ("HYPOXIA_UP"). The categories that were significantly enriched in the *Δtlo* mutant compared to WT in the presence of fluconazole included genes involved in amino acid transport and metabolism, iron and sulphur transport and metabolism, mitochondrial processes, oxidative shock and the opaque mating phenotype (Fig 3C). Although the ABC transporter encoding *CDR1* was found to be differentially expressed between the *Δtlo* mutant and the WT strain, both in the presence and absence of fluconazole (Figs 3 and 4), rhodamine 6G dye efflux assays showed no significant difference in drug efflux activity between these strains (Fig 1D). Additionally, the expression of *ERG11*, which is often overexpressed in resistant isolates (5,8), was unchanged between the WT strain and the *Δtlo* mutant both in the presence and absence of fluconazole. The mechanism responsible for the increase in MIC observed in the *Δtlo* mutant remains unclear and thus warrants further investigation.

## Differences in mitochondrial function between the WT and *Δtlo* mutant upon exposure to fluconazole

Gene set enrichment analysis of the transcriptomic data identified classes of genes involved in nutrient acquisition, cellular respiration and mitochondrial activity that were strongly induced by fluconazole only in the *Δtlo* mutant dataset. We further investigated the expression of these genes in the *Δtlo* mutant relative to WT in the presence and absence of fluconazole (Fig 4). Comparisons of the Log2 fold changes revealed that expression of these genes was largely

unchanged between the WT strain and *Δtlo* mutant in the absence of fluconazole, however in the presence of the drug, gene expression of related genes was significantly higher in the *Δtlo* mutant compared to the WT strain (Fig 4A). Comparison of the normalized mean expression values across all samples, suggests that the significant differences in mitochondrial-associated gene expression between the WT strain and *Δtlo* mutant in the presence of fluconazole is due to drug induced decreases in mitochondrial gene expression in the WT strain, whereas expression of these genes was largely unaffected by fluconazole in the *Δtlo* mutant (Fig 4B). The expression profile of the alternative oxidase *AOX2* was especially interesting, as its expression was downregulated in the WT strain upon exposure to fluconazole but was upregulated in the *Δtlo* mutant in the presence of the drug (Fig 4B).

## Differences in growth rates and oxygen consumption between the WT and *Δtlo* mutant upon exposure to fluconazole

To assess how differences in mitochondrial gene expression could affect the growth rates of the WT and *Δtlo* mutant strains, we monitored growth in RPMI over 24 h both in the presence and absence of fluconazole. The *Δtlo* mutant and the pENO-*TLOγ11* reintegrant displayed a prolonged lag phase but exhibited shorter doubling times compared to the WT strain and the pENO-*TLOα1* and pENO-*TLOβ2* reintegrants both in the presence and absence of fluconazole (Fig 5A & 5B). However, in the presence of the drug, the fluconazole tolerant *Δtlo* mutant and pENO-*TLOγ11* reintegrant reached significantly higher cell densities at the later time points (10–24 h) compared to the WT strain and the pENO-*TLOα1* and pENO-*TLOβ2* reintegrants (Fig 5B)

Next, we compared oxygen consumption rates, as a reflection of mitochondrial activity. These assays revealed that the level of oxygen consumption was lower in the *Δtlo* mutant relative to the WT strain during the exponential phase of growth. Reintegration of pENO-*TLOα1* and pENO-*TLOβ2* restored wild type levels of oxygen consumption, while reintegration of pENO-*TLOγ11* had no effect. This trend was even more pronounced in the presence of fluconazole and cyclosporine A, where it reached statistical significance (Fig 5C).

As the gene expression data indicated that exposure to fluconazole led to increased expression of the alternative oxidase *AOX2* in the *Δtlo* mutant, we hypothesized that the reduced oxygen consumption and fluconazole tolerance may be due to use of this alternative oxidase for respiration, which is known to consume less $O_2$ and yield fewer ATP than the complete respiratory chain. In order to determine if uncoupling of the respiratory chain could induce fluconazole tolerance, we examined the effects of electron transport chain inhibitors on fluconazole susceptibility (Fig 5D). The results show that the complex I inhibitor rotenone and the complex III inhibitor oligomycin A did not affect fluconazole tolerance in the WT strain and also completely inhibited tolerance in the *Δtlo* mutant. Although these data do not exclude a role for the alternative oxidase in tolerance, it demonstrates that inhibition of the respiratory chain cannot on its own induce fluconazole tolerance.

## Sterol biosynthesis pathway intermediates accumulate in the *Δtlo* mutant

Gene set enrichment analysis of the transcriptomic data suggests that sterol biosynthesis is disrupted in the *Δtlo* mutant. To examine this more closely the change in expression of each of the ergosterol biosynthesis (*ERG*) genes between the WT strain and the *Δtlo* mutant was compared directly both in the presence and absence of fluconazole (Fig 6A). The results show that in the absence of fluconazole, key genes such as *ERG3*, *ERG6*, *ERG2*, *ERG25* and *ERG251*, were significantly downregulated in the *Δtlo* mutant. Upon exposure to fluconazole, *ERG* gene expression was induced in both the WT strain and the *Δtlo* mutant, however this fluconazole-

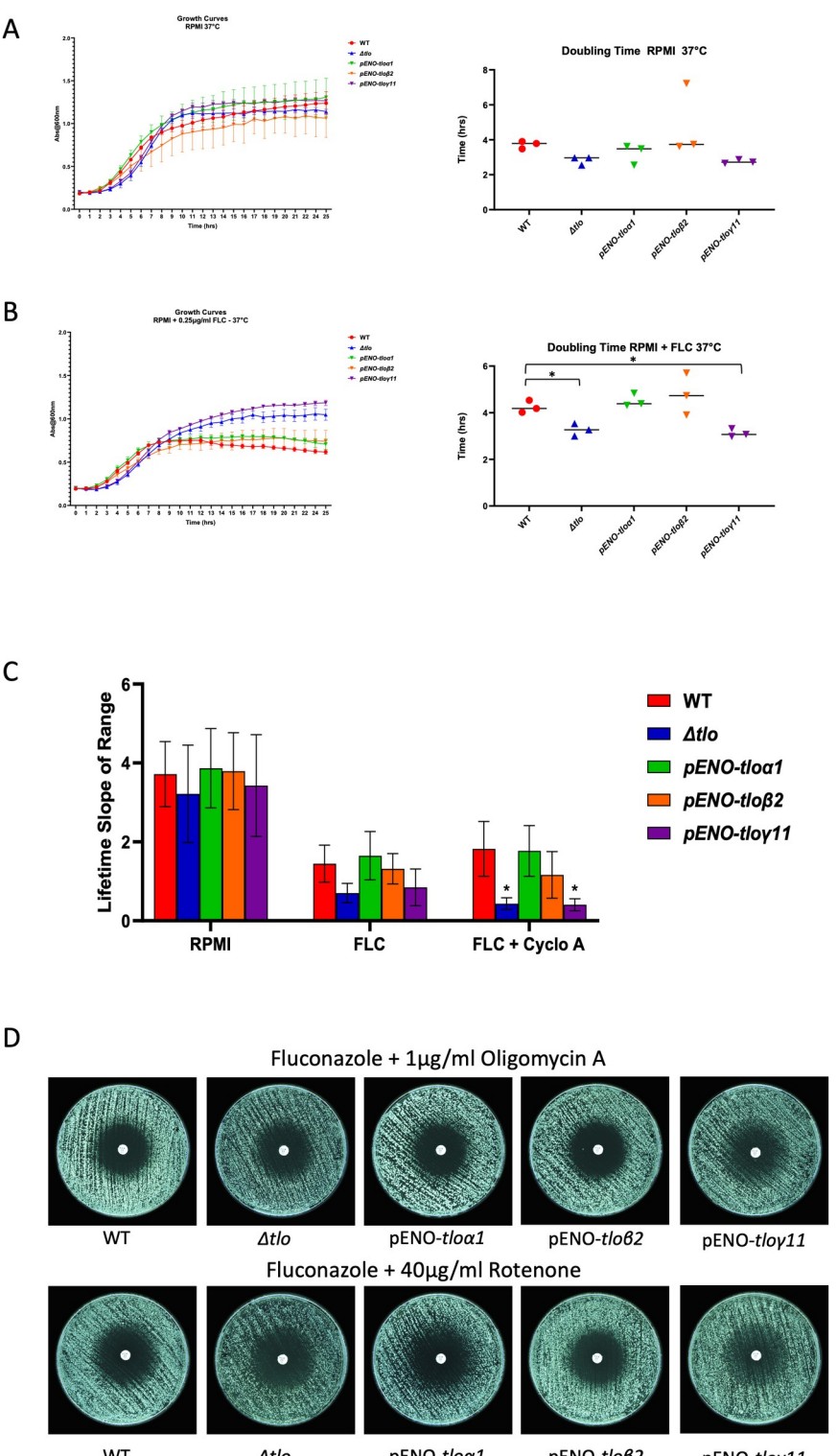

**Fig 5. Growth and mitochondrial activity and of the WT strain, the *Δtlo* mutant and the *TLO* reintegrants in the presence and absence of fluconazole.** Growth and mid-exponential phase doubling time (3–9 h) of the WT strain, the *Δtlo* mutant and *TLO* reintegrants in RPMI liquid medium over 24hrs at 37°C, both in the absence (A) and the presence of fluconazole (0.25 μg/ml) (B). Oxygen consumption rate, a measure of mitochondrial activity, of the WT strain, *Δtlo* mutant and *TLO* reintegrant strains in the absence of any treatment (RPMI), in the presence of fluconazole (FLC) and in the presence of fluconazole and cyclosporine A FLC+CycloA). Asterisks indicate the statistically

significant differences between the WT strain and mutant and reintegrant strains, * = p-value < 0.05 (C). Fluconazole disk diffusion assays comparing the sensitivity of WT, the *Δtlo* mutant, and the *TLO* reintegrants in the presence electron transport chain inhibitors oligomycin A and rotenone (D).

induced upregulation was more pronounced in the WT strain. Therefore, many of the *ERG* genes had lower levels of expression in the *Δtlo* mutant than the WT strain when exposed to fluconazole (Fig 6B). The most notable differences between the WT strain and *Δtlo* mutant upon exposure to fluconazole was the expression of *ERG251*, *ERG6* and the transcriptional regulator *UPC2*. Analysis of RNA-sequence data described in our previous work [22], where strains were incubated in nutrient-rich YPD liquid medium (without fluconazole) revealed that the reintegration of pENO-*TLOα1*, and pENO-*TLOβ2* restored the expression of the *ERG* genes to levels comparable with the WT strain. However, there were some differences in the relative ability of *TLOα1* and *TLOβ2* to restore expression levels of individual *ERG* genes, suggesting the encoded proteins may differ in their functionality. Reintegration of pENO-*TLOγ11* did not restore *ERG* gene expression to WT levels (Fig 6C).

To examine how these changes in *ERG* gene expression affected the sterol content of the WT strain, the *Δtlo* mutant and the pENO-*TLO* reintegrated strains, membrane sterol composition analysis was carried out using mass spectrometry, both in the absence and presence of sub-inhibitory concentrations of fluconazole (0.25 μg/ml). The results reveal that in the absence of fluconazole, the percentage of ergosterol was significantly higher in the WT strain membranes compared to the *Δtlo* mutant (Fig 7A). Complementation of the mutant with pENO-*TLOα1* and pENO-*TLOβ2* restored the ergosterol content to WT levels, however reintegration of *TLOγ11* had no effect. The presence of fluconazole induced a marked reduction in the level of ergosterol in the WT strain, and significant decreases in ergosterol levels in the *TLOα1* and *TLOβ2* reintegrants. The presence of fluconazole had no effect on the levels of ergosterol in the *Δtlo* mutant and *TLOγ11* reintegrant (Fig 7A). The analysis of membrane sterol composition also revealed the accumulation of several sterol intermediates in the *Δtlo* mutant in the absence of fluconazole, these included lanosterol, 4,4-dimethyl cholesta-8,24-dienol, zymosterol, fecosterol (ergosta-8,24(28)-dienol), episterol (ergosta-7,24(28)-dienol), ergosta-7-enol, and ergosta-5,7-dienol. These accumulations correlate with reduced expression of genes encoding the specific enzymes required for their metabolism, including *ERG6*, *ERG25 (ERG251)* and *ERG3* compared to the WT strain (S1 Fig). Some of these changes (e.g. episterol accumulation) were alleviated upon the reintegration of pENO-*TLOα1* and pENO-*TLOβ2*, but not by pENO-*TLOγ11* (Fig 7B).

Exposure of the strains to a subinhibitory concentration of fluconazole led to increased accumulation of lanosterol with the *Δtlo* mutant and WT exhibiting reduced accumulation of sterol intermediates, such as 4,4-dimethyl cholesta-8,24-dienol, zymosterol, fecosterol, episterol, ergosta-7-enol, ergosta-7,22 dienol and ergosta-5,7-dienol (Fig 7B). Upon exposure to fluconazole, the expression of *ERG25 (ERG251)* and *ERG3* was partially restored in the *Δtlo* mutant, leading to the alleviation of the build-up of most sterol intermediates.

## Overexpression of individual ergosterol biosynthesis genes in the *Δtlo* mutant

In order to investigate if reduced expression of specific *ERG* genes might play a role in the development of antifungal tolerance, their expression was artificially increased in the *Δtlo* mutant, verified by qRT-PCR and their fluconazole susceptibility assessed. The results show that increasing the expression of individual ergosterol biosynthesis genes had no effect on the fluconazole susceptibility of the *Δtlo* mutant, which remained highly tolerant to the drug

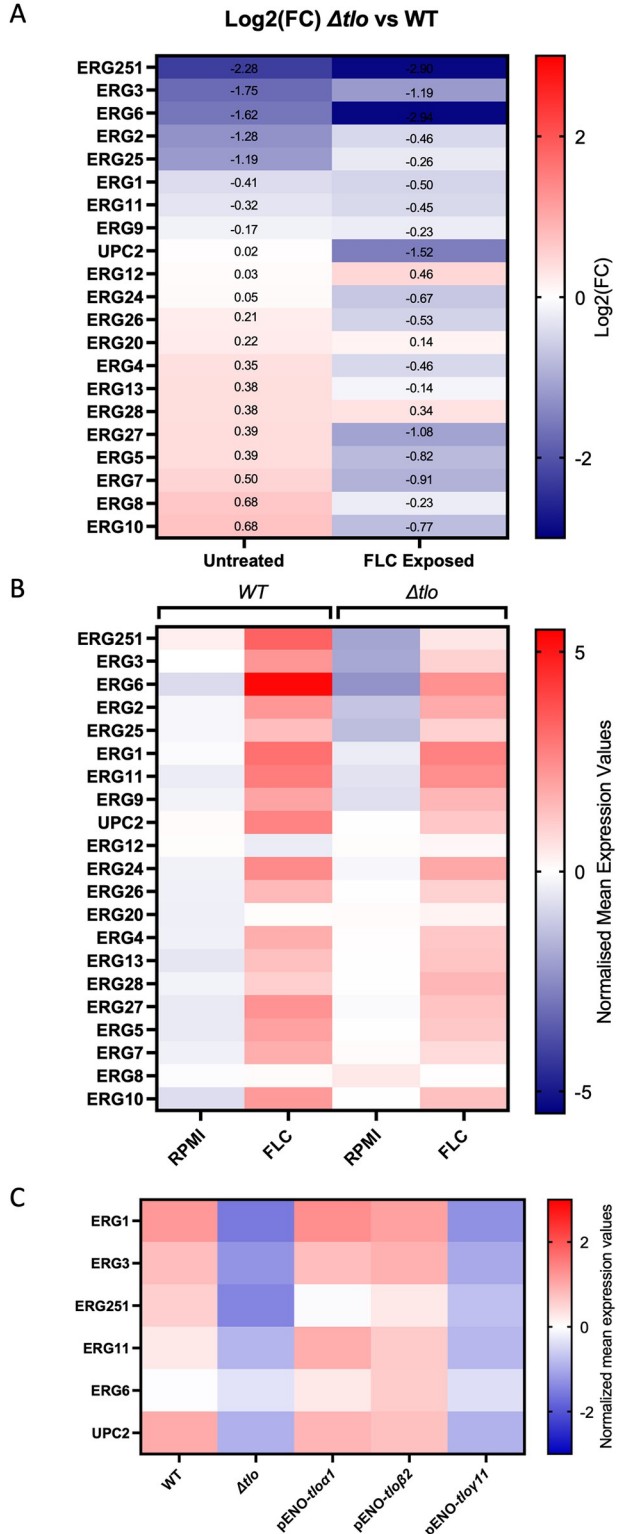

**Fig 6. Differential expression of each of the ergosterol biosynthesis genes between the WT strain and the *Δtlo* mutant in the presence and absence of fluconazole.** Log2 fold change differences in the expression of each ergosterol biosynthesis gene in the *Δtlo* mutant compared to WT are shown both in the absence (untreated) and the presence (FLC Exposed) of fluconazole (A). The normalized mean expression values of each ergosterol biosynthesis gene in untreated and fluconazole exposed samples in both the WT strain and the *Δtlo* mutant (B) The normalized mean

expression values of key *ERG* genes in WT, *Δtlo* mutant and the *TLO* reintegrant strains when grown to mid-exponential phase in nutrient rich YPD medium at 37˚C (C).

(S2 Fig). Sterol analysis revealed that increasing the expression of *ERG1*, *ERG3* and *ERG6* partially restored WT levels of ergosterol in the absence of fluconazole, however, there were no significant differences in the levels of ergosterol between the WT strain, the *Δtlo* mutant and any of the *ERG* gene overexpression mutants in the presence of the drug (S2B Fig). Sterol composition analysis also provided confirmation of successful overexpression of individual ergosterol biosynthesis genes (S2C Fig), as for example, accumulation of ergosta-7 enol in the *Δtlo* mutant was reversed upon upregulation of *ERG3*. Similarly, accumulation of zymosterol in the *Δtlo* mutant was reversed upon upregulation of *ERG6*.

## Discussion

The massive expansion of the *TLO* gene family is a unique feature of *Candida albicans* among fungal species and eukaryotes in general. In order to investigate the role of this large family in the success of *C. albicans* as coloniser and pathogen of humans we recently used CRISPR-Cas9 mutagenesis to generate a *C. albicans Δtlo* mutant, in which all 14 paralogs were simultaneously deleted [22]. Analysis of this mutant revealed defects in a wide variety of cellular processes, including metabolic pathways, morphology, stress responses and virulence. These defects were alleviated (sometimes to varying extent) when the *Δtlo* mutation was complemented by reintroducing representatives of the alpha- and beta-*TLO* gene clades, but not by representatives of the gamma-clade. In the current study we further investigated the phenotype of *Δtlo* mutant by investigating if deletion of the *TLO* family affects how *C. albicans* responds to the commonly used antifungal drug fluconazole.

The comparative fluconazole sensitivity of the WT, the *Δtlo* mutant and reintegrants in which clade-representative *TLO* genes were reintroduced into the mutant was first assessed using disk diffusion and Etest sensitivity assays (Fig 1). The results show a reduction in the sensitivity of the *Δtlo* mutant to fluconazole compared to the WT strain, which was restored upon reintegration of *TLOα1* and *TLOβ2*, but not *TLOγ11* (Fig 1). In order to distinguish between tolerance and resistance, similar experiments were performed in the presence of cyclosporine A, an inhibitor of antifungal tolerance through inhibition of the calcineurin pathway [10, 21]. This revealed clearer zones of inhibition suggesting that the *Δtlo* mutant is tolerant of fluconazole under these conditions. A detectable increase in MIC also suggests that the mutant is somewhat less susceptible than WT (Fig 1). Therefore, the loss of sensitivity observed in the *Δtlo* mutant is likely due to the combined effects of tolerance and resistance mechanisms.

In order to investigate the tolerance phenotype of the mutant further we conducted comparative transcriptomic profiling and subsequent Gene Set Enrichment Analysis (GSEA) of the WT strain and the *Δtlo* mutant in the absence and presence of fluconazole (Figs 2 and 3). Among the most prominent differences between the WT strain and *Δtlo* mutant was the expression of genes involved in metabolic processes, such as mitochondrial function and energy metabolism. We observed increased expression of genes encoding proteins involved in mitochondrial processes and cellular respiration in the *Δtlo* mutant, relative to the WT strain, in the presence of fluconazole (Fig 4). Comparison of the normalized mean expression values across all samples revealed that this difference in expression was due to fluconazole induced decreases in mitochondrial gene expression in the WT strain, while the expression of the same genes in the *Δtlo* mutant was largely unaffected by fluconazole (Fig 4B). Despite the higher levels of mitochondrial gene expression in the *Δtlo* mutant, mitochondrial oxygen consumption

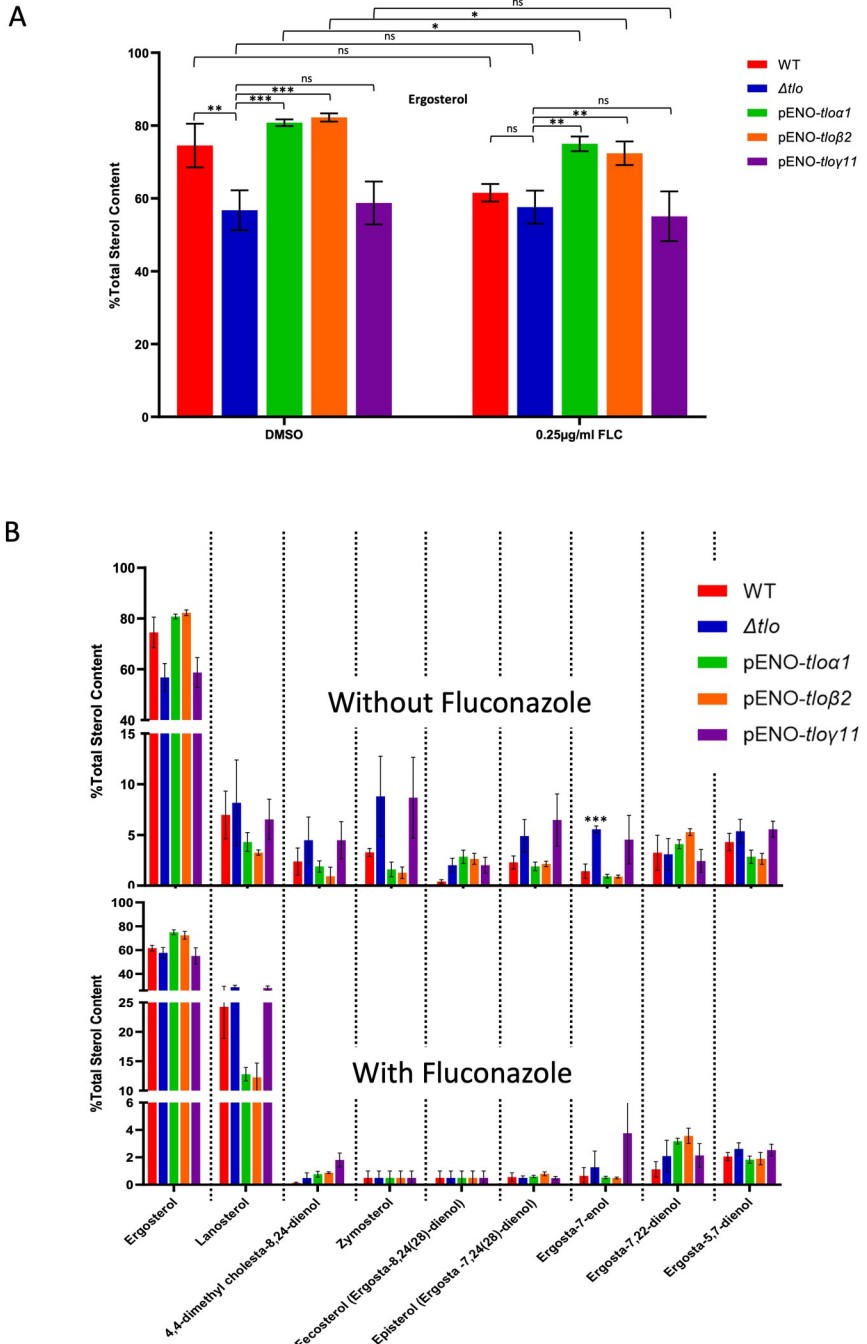

**Fig 7. Sterol composition analysis of the WT strain and the *Δtlo* mutant in the presence and absence of fluconazole.** Ergosterol content in the cell membranes of the WT strain, *Δtlo* mutant and *TLO* reintegrants. Asterisks indicate statistical significance; where * = p-value <0.05, ** = p-value <0.01 and *** = p-value <0.001; square brackets indicate the conditions being compared (A). Relative abundance of sterol intermediates of the ergosterol biosynthesis pathway in the WT strain, *Δtlo* mutant, and *TLO* reintegrants, both in the absence and the presence of fluconazole. Asterisk indicates statistical significance, where *** = p-value <0.001, between the WT strain and *Δtlo* mutant (B).

assays revealed that the *Δtlo* mutant has reduced oxygen consumption and this was highly significant in the presence of fluconazole (Fig 5C). Despite the reduced metabolic activity, the *Δtlo* mutant exhibited persistent growth in the presence of fluconazole compared to the WT

strain and the *TLOα1* and *TLOβ2* reintegrants, consistent with the azole tolerant phenotype (Fig 5B).

Taken together these data suggest that the *Δtlo* mutant may be using the alternative respiratory pathway. This was supported by the expression profile of the alternative oxidase *AOX2*, whose expression was induced in the *Δtlo* mutant upon exposure to fluconazole but reduced in the WT strain under the same conditions, leading to a an approximately 5-fold Log2 change in expression (p-value <0.001). The alternative oxidase pathway consumes less oxygen and dramatically reduces the energy (ATP) yield of respiration, since electrons flowing to alternative oxidase bypass the proton pumping complexes III and IV [41]. In order to determine if tolerance in the *Δtlo* involves uncoupling of the respiratory chain, we exposed *C. albicans* to the complex I inhibitor, rotenone, and the complex III inhibitor, oligomycin A in the presence of fluconazole (Fig 5D). The results show that wild-type *C. albicans* did not develop tolerance in the presence of both electron transport chain inhibitors, suggesting that the uncoupling of oxidative phosphorylation from ATP synthesis is not solely responsible for the acquisition of fluconazole tolerance by the *Δtlo* mutant. However, this finding does not negate the potential relevance of increased *AOX2* expression in the development of fluconazole tolerance, as the increased expression of the alternative oxidase has previously been shown to reduce susceptibility of *C. albicans* to fluconazole, owing to its ability to neutralise reactive oxygen species (ROS) [42]. Increased expression of *AOX2* may also be related to changes in ergosterol biosynthesis, as the alternative oxidase was also upregulated in an *ERG251* mutant in response to azoles [43]. Together these data suggest that the alternative oxidase Aox2 may play a central role in the development of fluconazole tolerance in the *Δtlo* mutant.

The transcriptomic analysis also revealed differential expression of several genes related to oxidative stress responses between the *Δtlo* mutant and the WT strain upon exposure to fluconazole (Figs 3 and 4), which is likely the result of impaired oxidative phosphorylation and cellular respiration in the *Δtlo* mutant leading to the generation of ROS [44, 45]. This finding is of added importance in the context of antifungal tolerance, as altered mitochondrial activity has previously been linked to changes in antifungal susceptibility as a result of changes in plasma membrane architecture and permeability mediated through changes in phospholipid, sphingolipid and sterol biosynthesis, as well as activation of PDR signalling [15, 20].

The transcriptomic data also show that the *TLO* genes affect the expression of genes responsible for ergosterol biosynthesis (*ERG*). Expression of the *ERG* genes and the *ERG* gene regulator *UPC2* was higher in the WT strain compared to the *Δtlo* mutant. Fluconazole induced *ERG* gene expression was observed in both WT and the *Δtlo* mutant, however the level of induction was higher in WT (Fig 6A and 6B). This suggests that the *TLO* genes are required for efficient expression of the *ERG* genes, particularly in the presence of fluconazole.

Analysis of sterol profiles in the mutant showed that defective expression of the *ERG* genes is associated with reduced levels of ergosterol in the membrane and increased levels of precursor sterols, particularly lanosterol. In the presence of fluconazole, several of the sterol intermediates were less abundant in the *Δtlo* mutant sterol profile, which may be due to fluconazole induced *ERG* gene expression. Interestingly, *ERG6* and *ERG251* were the most differentially expressed genes of the ergosterol biosynthesis pathway between the WT strain and *Δtlo* mutant upon exposure to fluconazole (Fig 6A and 6B). This difference is especially interesting as both genes have recently been implicated in the development of antifungal tolerance in *Candida* species [43, 46, 47]. It is also worth noting that expression of both *ERG6* and *ERG251* was higher upon reintegration of pENO-*TLOβ2* than upon reintegration of pENO-*TLOα1*, and this was associated with greater reversal of the tolerance phenotype by pENO-*TLOβ2*, demonstrating *TLO*-specific effects on ergosterol biosynthesis and fluconazole susceptibility. We sought to further investigate the potential involvement of the differentially expressed *ERG*

genes by restoring the expression of specific *ERG* genes in the *Δtlo* background by placing them under the control of the strong enolase (*ENO*) gene promoter (S2 Fig). Although results from qRT-PCR and sterol analysis confirmed successful upregulation of the individual *ERG* genes in the *Δtlo* mutant background, this did not have any effect on fluconazole tolerance. It is possible that upregulation of the individual *ERG* genes, including *ERG251*, had not reached the sufficient threshold required to elicit any observable effect on the tolerance phenotype, or more likely that changes in fluconazole tolerance would require increased expression of multiple genes in the biosynthetic pathway. The relationship between Erg251 and fluconazole susceptibility is particularly interesting, as there currently exists contradictory evidence in the literature pertaining to its effect. For example, one study demonstrated that inhibition of Erg251 by way of a small molecule inhibitor potentiated the effects of fluconazole, causing it to become fungicidal against *C. albicans* [46], while another study demonstrated that transposon mutagenesis of *ERG251* in a haploid *C. albicans* strain resulted in decreased fluconazole susceptibility [48]. Most recently, a third study has shown that loss of function point mutations in *ERG251* arise during adaption of *C. albicans* to antifungal stress, which lead to increased azole tolerance [43]. In addition to this, transcriptomic analysis of an *erg251Δ* mutant in the study by Zhou *et al.* revealed differences in the expression in nutrient acquisition and stress response genes, which were similar to the differences in gene expression profile observed in the *Δtlo* mutant in the present study [43]. In this way, our *Δtlo* mutant very closely phenocopies the *ERG251* mutant of the Zhou *et al.* study, suggesting that the reduced expression or activity of Erg251 in the *Δtlo* mutant relative to the WT strain in the presence of fluconazole may be a contributing factor to the changes observed in metabolic processes and increased fluconazole tolerance observed.

Sterol analysis of the *ERG* gene overexpression mutants also provided additional insight into the ergosterol biosynthesis pathway of the *Δtlo* mutant, as overexpression of *ERG11*, but not *ERG6*, relieved the accumulation of lanosterol in the *Δtlo* mutant. This suggests that the *Δtlo* mutant is likely utilising the alternative sterol biosynthesis pathway, leading to the accumulation of toxic sterol intermediates [13]. Although sterol intermediates are accumulated in the *Δtlo* mutant in the absence of fluconazole, these accumulations are largely reversed upon exposure of the mutant to fluconazole. Specifically, the *Δtlo* mutant displayed reduced abundance of ergosterol and the accumulation of several toxic sterol intermediates, compared to the WT strain. This suggests that due to the absence of Tlo proteins, the *Δtlo* mutant has constitutively altered its sterol profile in a manner that resembles a response to fluconazole exposure, even before the drug is present, which may explain why the *Δtlo* mutant displays constitutively lower levels of sensitivity to the drug than the WT strain.

To conclude, we propose that the *TLO* gene family, through their role in controlling the expression of genes required for metabolism, mitochondrial function and ergosterol biosynthesis, play an important role in how *C. albicans* responds to fluconazole, in particular to how this species exhibits tolerance of this drug. Further studies to identify how the Tlo proteins are implicated in tolerance will include gene single gene ChIP-Seq and co-IP experiments as well as experiments involving other members of the alpha- and gamma-*TLO* clades to determine if other members of these clades differ in their contribution to the responses of *C. albicans* to fluconazole.

## Supporting information

**S1 Fig. Schematic diagrams of ergosterol biosynthesis pathways in the absence and in the presence of subinhibitory concentrations of fluconazole.** Sterol intermediates in bold signify their accumulation, the Erg proteins highlighted in blue signify the downregulation of the encoding *ERG* gene in the *Δtlo* mutant relative to the WT strain. In the absence of the drug,

several toxic intermediates of the ergosterol biosynthesis pathway accumulated in the *Δtlo* mutant, such as lanosterol, 4,4-dimethyl cholesta-8, 14, 24-trienol, zymosterol, fecosterol and episterol. The accumulation of these intermediates coincides with the down regulation of the *ERG* genes encoding enzymes that catalyses their conversion, which are highlighted in blue (A). In the presence of the fluconazole, the build-up of these toxic sterol intermediates in the *Δtlo* mutant is alleviated, although differential *ERG6* expression contributes to the build-up of lanosterol in the *Δtlo* mutant (B).
(TIF)

**S2 Fig. Fluconazole sensitivity and sterol composition of ergosterol gene overexpression mutants in the *Δtlo* background.** The sensitivity of several ergosterol gene overexpression mutants in the *Δtlo* mutant background was tested using fluconazole disk diffusion assays (A). Ergosterol levels of the WT strain, *Δtlo* mutant and *ERG* gene overexpression mutants in the presence and absence of fluconazole (B). Sterol composition of the WT strain, the *Δtlo* mutant and the *ERG* gene overexpression mutants in the absence and the presence of fluconazole (C). Asterisks indicate statistical significance; where * = p-value <0.05, ** = p-value <0.01 and *** = p-value <0.001 between the WT strain and the mutants tested.
(TIF)

**S1 Table. Genotypes of strains used in the study.**
(XLSX)

**S2 Table. Oligonucleotides used in the study.**
(XLSX)

**S3 Table. Genes differentially expressed between WT and *tlo* null deletion mutant cultured on solid RPMI medium for 48 h at 37˚C in the absence of fluconazole.**
(XLSX)

**S4 Table. Genes differentially expressed between WT and *tlo* null deletion mutant cultured on solid RPMI medium for 48h at 37˚C in the presence of fluconazole.**
(XLSX)

## Author Contributions

**Conceptualization:** Gary P. Moran, Derek J. Sullivan.

**Data curation:** James O'Connor-Moneley, Gary P. Moran.

**Formal analysis:** James O'Connor-Moneley, Jessica Fletcher, Josie Parker, Gary P. Moran.

**Funding acquisition:** Gary P. Moran, Derek J. Sullivan.

**Investigation:** James O'Connor-Moneley, Jessica Fletcher, Cody Bean, Josie Parker, Steven L. Kelly, Gary P. Moran.

**Methodology:** James O'Connor-Moneley, Gary P. Moran, Derek J. Sullivan.

**Project administration:** Gary P. Moran, Derek J. Sullivan.

**Resources:** Gary P. Moran.

**Supervision:** Gary P. Moran, Derek J. Sullivan.

**Validation:** Gary P. Moran, Derek J. Sullivan.

**Visualization:** Gary P. Moran, Derek J. Sullivan.

**Writing – original draft:** James O'Connor-Moneley, Derek J. Sullivan.

**Writing – review & editing:** James O'Connor-Moneley, Gary P. Moran, Derek J. Sullivan.

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
