## [Decision Letter · Decision Letter 0]

23 Jul 2024

PONE-D-24-27239Deletion of the Candida albicans TLO gene family results in alterations in membrane sterol composition and fluconazole tolerancePLOS ONE Dear Dr. Sullivan,

Thank you for submitting your manuscript to PLOS ONE. After careful consideration, we feel that it has merit but does not fully meet PLOS ONE’s publication criteria as it currently stands. Therefore, we invite you to submit a revised version of the manuscript that addresses the points raised during the review process.

We look forward to receiving your revised manuscript.

Kind regards,

Rajendra Upadhya

Academic Editor

PLOS ONE

Journal Requirements:

"The authors would like to acknowledge support from Science Foundation Ireland (SFI, Grant No. 19/FFP/6422 [DJS & GPM]) and the Dublin Dental University Hospital. The funders did not play any role in study design, data collection, analysis or in the decision to publish."

Please note that funding information should not appear in the Acknowledgments section or other areas of your manuscript. We will only publish funding information present in the Funding Statement section of the online submission form. Please remove any funding-related text from the manuscript.

3. Please note that your Data Availability Statement is currently missing the direct link to access each database. If your manuscript is accepted for publication, you will be asked to provide these details on a very short timeline. We therefore suggest that you provide this information now, though we will not hold up the peer review process if you are unable.

5. Please include all captions for your Supporting Information files at the end of your manuscript, and update any in-text citations to match accordingly. Please see our Supporting Information guidelines for more information: http://journals.plos.org/plosone/s/supporting-information. 

**Additional Editor Comments:**

Dear Drs. Sullivan and Moran, 

We appreciate your submission of the work to PLOS ONE. This manuscript has undergone a thorough examination by two reviewers, both of whom appreciated the work and acknowledged that the results of the manuscript offer a valuable dataset that could potentially contribute to our understanding of the role of the *TLO* gene family in azole tolerance. I believe that addressing a few minor concerns identified by one of the reviewers will enhance the clarity of the work for the readers. 

Reviewers' comments:

Reviewer's Responses to Questions

**Comments to the Author**

1. Is the manuscript technically sound, and do the data support the conclusions?

Reviewer #1: Yes

Reviewer #2: Partly

2. Has the statistical analysis been performed appropriately and rigorously? 

Reviewer #1: Yes

Reviewer #2: I Don't Know

3. Have the authors made all data underlying the findings in their manuscript fully available?

Reviewer #1: Yes

Reviewer #2: Yes

4. Is the manuscript presented in an intelligible fashion and written in standard English?

Reviewer #1: Yes

Reviewer #2: Yes

5. Review Comments to the Author

Reviewer #1: The intention of the authors of the manuscript was to investigate the influence of the TLO gene family in Candida albicans on its resistance to fluconazole. The publication showed that the deltatlo mutant is more resistant to fluconazole than the initial strain (WT). The next stage of the research was to obtain transcriptomes of the tested strains growing without and with fluconazole and to analyse gene expression. Based on the transcriptome analysis, the authors selected possible mechanisms of C. albicans resistance to fluconazole and used the qPCR method to examine the expression of the selected genes and the activities of the proteins encoded by them. It can be concluded that the research methods adopted by the authors is absolutely correct. As is often the case, the results obtained did not provide a clear answer to the question of what mechanism is responsible for the resistance of deltatlo mutants to fluconazole. The effect appears to occur through pleiotropic changes in the cell, including perhaps as yet unknown mechanisms.

The work submitted for publication indicates a number of issues whose explanation should be continued and deepened. The ms is therefore rather an introduction to further research. However, this type of work undoubtedly contributes to the development of science in the field of fungal resistance to azoles and that's why I recommend ms to be published in Plos One.

Line 608-612 – the authors should write what they want to present in a different way, because now it seems that these two sentences contradict each other.

Reviewer #2: This article describes the characterization of a Candida albicans mutant deleted for all members of the TLO gene family, encoding a subunit of the mediator, regarding fluconazole tolerance. The authors use transcript profiling comparison to uncover the molecular basis for drug tolerance in the mutant and show a link with respiration; they also analyze cell membrane composition in parallel with expression studies on individual ERG genes.

Overall, the data presented here are rather convincing even if they do not uncover a major pathway linking TTLO genes and Fluconazole tolerance. However, I have several concerns that need to be addressed:

In the result section, the authors should mention that the tlo mutant displays some resistance toward fluconazole, as experiments clearly show an increase of MIC (from 0.25 to 0.75 µg/ml).

The KO mutant is complemented by either one of 3 genes, alpha1, beta2 or gamma11, representative of the three gene clades. A discussion about their similarities is missing. Moreover, to my knowledge, no NLS could be predicted for TLO11. Maybe another gene from that clade could have been used, or the grounds for the choice discussed.

In the discussion, the authors suggest that the observed phenotype could be linked with a reduced expression/activity of Erg251. Yet, overexpressing ERG251 in the tlo KO had no impact, suggesting the involvement of other proteins.

There is a discrepancy between the results section and legend of figure 6, which states that cells were grown in YPD. Which is correct?

Why have the authors used the CDG C. albicans Assembly 21 to analyze the RNA-sequencing data, and yet use the systematic names of Assembly 22?

The genotypes should be properly written at least in the parental strain, and each new mutant also described, including selection markers. The med3 KO mutant is not in the table. And please use either gene names (eg TLO1) or phylogenetic names (TLOalpha1), and be consistent throughout the article, including figures and legends. Same for strain names DSY, and not Dsy.

Fig1A and 1C: replace “RPMI only” by “Fluconazole”, and “RPMI containing …” by “Fluconazole + …”, as in Figure 5.

S Figure 2: ergosta-5,7-dienol should be in bold.

Typos: line 140 and Fig 4A (Fluconaozle); line 178 (50 mls); lines 245, 246, 249 (an MIC); lines 296, 332 (preformed); line 452 (cholestra).

Check syntax line 140-141, line 164-165.

6. PLOS authors have the option to publish the peer review history of their article (what does this mean?). If published, this will include your full peer review and any attached files.

Reviewer #1: No

Reviewer #2: No

---

## [Author Response · Author response to Decision Letter 0]

26 Jul 2024

Academic Editor/Journal Requirements:

1. PLOS ONE style requirements

We have reviewed the document and made some minor adjustments in line with the recommended templates. All submitted files have been named as recommended by PLOS ONE.

2. Acknowledgements

We have removed any reference to funding in the text. Therefore, there is no longer a need for a separate acknowledgements section, and this has now been deleted (see lines 640-643).

3. Data Availability Statement

The URL which links directly to the RNA-sequencing data submitted to the National Centre for Biotechnology Information (NCBI) sequence read archive has been added to the Data Availability Statement as requested.

4. Abstract

The discrepancy between the two abstracts was the result of the online submission form being unable to recognise Greek lettering. We have now removed these from both abstracts, and they should now be identical.

5. Supporting Information file captions.

We had omitted the captions for the Tables included as Supporting Information and these have now been added (see lines 802-810).

6. Reference list

The reference list is complete and correct. No changes have been made from the original submission.

Reviewer # 1

The first sentence refers specifically to the ∆tlo mutant, while the second sentence was originally describing the ERG-overexpressing strains. However, to remove any ambiguity we have deleted the second sentence (see lines 621-625), as this is not essential.

Reviewer # 2

1. Resistance phenotype

Deletion of the TLO gene family results in an increase in tolerance and resistance of C. albicans to fluconazole. We felt that we had highlighted this in the original text, but we have reiterated the increase in resistance (see lines 252-253) to ensure this is now clearer.

2. Comparison of the TLO clades

We have included additional text (see lines 102-105) in the Introduction highlighting differences that we have previously observed between the gamma-clade genes and the apha- and beta-genes. In a previous study (reference [22]) we compared TLOy11 and another gamma gene, TLOy5, and both were found to have identical effects, suggesting that the choice of TLOy11 as a gamma-clade representative is sound. We have also added a sentence at the end of the Discussion (see lines 634-638) stating that future experiments will include additional members of each of the alpha- and gamma-TLO clades to see if there is intra-clade variability in Tlo protein functionality. As there is only one member of the beta clade there is no need for any further analysis of this clade.

3. ERG251

We agree with the reviewer (and with Reviewer #1), that the tolerance phenotype is almost certainly pleiotropic and that additional proteins are likely to be implicated. We had originally included this in the text, but have specifically referenced ERG251 in this context (see line 597-598) to emphasise this point.

4. Figure 6

The data described in Fig 6C are based on the analysis of a dataset from a previously published study (i.e. reference [22]), in which similar RNA-seq experiments were conducted on strains grown in liquid YPD medium. This has now been explicitly highlighted in the text (see lines 432-434)

5. Genome assemblies.

We have corrected the text (see line 165) to refer to the CGD C. albicans Genome Assembly 22, not Assembly 21, as originally stated.

6. Genotypes

S1 Table, the supplementary table detailing the genotypes of strains used in the study, has been amended, to include the addition of the med3 deletion mutant and selection markers. Gene and strain names are now used consistently throughout the text and figures as suggested.

7. Figures

Fig 1A and 1C have been amended, however, S2 Fig: ergosta-5,7-dienol is already in bold 

8. Typos and syntax

All suggested corrections have been made, although we believe that “an MIC” is correct.

---

## [Editor Report · Decision Letter 1]

29 Jul 2024

Deletion of the Candida albicansTLO gene family results in alterations in membrane sterol composition and fluconazole tolerance

PONE-D-24-27239R1

Dear Drs. Sullivan and Moran, 

We’re pleased to inform you that your revised manuscript has been judged scientifically suitable for publication and will be formally accepted for publication once it meets all outstanding technical requirements.

Kind regards,

Rajendra Upadhya

Academic Editor

PLOS ONE
---

## [Editor Report · Acceptance letter]

1 Aug 2024

PONE-D-24-27239R1 

PLOS ONE

Dear Dr. Sullivan, 

I'm pleased to inform you that your manuscript has been deemed suitable for publication in PLOS ONE. Congratulations! Your manuscript is now being handed over to our production team.

Kind regards, 

on behalf of

Dr. Rajendra Upadhya 

Academic Editor

PLOS ONE